# Frailty Risk Prediction Model among Older Adults: A Chinese Nation-Wide Cross-Sectional Study

**DOI:** 10.3390/ijerph19148410

**Published:** 2022-07-09

**Authors:** Siying Li, Wenye Fan, Boya Zhu, Chao Ma, Xiaodong Tan, Yaohua Gu

**Affiliations:** 1School of Public Health, Wuhan University, Wuhan 430071, China; lisiying@whu.edu.cn (S.L.); 2021283050081@whu.edu.cn (W.F.); 2021283050060@whu.edu.cn (B.Z.); 2021283050105@whu.edu.cn (C.M.); 2School of Nursing, Wuhan University, Wuhan 430071, China

**Keywords:** frailty, risk factor, prediction model, social support, lifestyle

## Abstract

Objectives: Numerous studies have been performed on frailty, but rarely do studies explore the integrated impact of socio-demographic, behavioural and social support factors on frailty. This study aims to establish a comprehensive frailty risk prediction model including multiple risk factors. Methods: The 2018 wave of the Chinese Longevity and Health Longitudinal Survey was used. Univariate and multivariate logistic regressions were performed to identify the relationship between frailty and multiple risk factors and establish the frailty risk prediction model. A nomogram was utilized to illustrate the prediction model. The area under the receiver operating characteristic curve (AUC), Hosmer–Lemeshow test and calibration curve were used to appraise the prediction model. Results: Variables from socio-demographic, social support and behavioural dimensions were included in the final frailty risk prediction model. Risk factors include older age, working as professionals and technicians before 60 years old, poor economic condition and poor oral hygiene. Protective factors include eating rice as a staple food, regular exercise, having a spouse as the first person to share thoughts with, doing physical examination once a year and not needing a caregiver when ill. The AUC (0.881), Hosmer–Lemeshow test (*p* = 0.618), and calibration curve showed that the risk prediction model was valid. Conclusion: Risk factors from socio-demographic, behavioural and social support dimensions had a comprehensive effect on frailty, further supporting that a comprehensive and individualized intervention is necessary to prevent frailty.

## 1. Introduction

Frailty refers to the decline in functions of multiple body systems, making the body more vulnerable to stress [1]. With the development of medical science and the extension of human life expectancy, the population is ageing rapidly worldwide. Many chronic conditions that are associated with older adults, such as frailty, have become increasingly significant public health problems [2]. Chronic frailty harms physical health, reduces quality of life [3] and increases medical and health expenditure, leading to a heavy economic burden on families and society [4]. Frailty is an adverse health condition that occurs with human age, but unlike ageing, the frailty process can be stopped or even reversed if appropriate interventions and preventive measures are implemented [1].

So far, thorough and systematic frailty studies have been implemented to explore how risk factors from multiple dimensions affect frailty. Previous studies have confirmed that socio-demographic factors such as sex, age, income [1], marital status [5], education [6] and residence [7]; behavioural factors such as smoking [8], drinking [9], exercise [10] and diet [11]; and social support factors [12,13] such as having a caregiver [14], engaging in community service [15] and social insurance are all associated with frailty. However, rarely do studies explore the integrated impact of those risk factors on frailty. In addition, most frailty prediction models have been carried out in developed countries [16]. For example, a study from Tokyo developed a predictive frailty model that was based on socio-demographic, medical, behavioural and subjective factors [17]. A study from France developed the FRAGIRE tool to assess the frailty risk in older adults [18]. Due to significant differences in socio-demographic characteristics, behaviour, and social environment between developed and developing countries, the frailty prediction models are leagues apart [19]. What remains unknown is how socio-demographic, behavioural, and social support factors influence and predict frailty together in developing countries. Hence, there is a need to establish a frailty risk prediction model for Chinese older adults, the largest ageing population in the world (approximately 164,487 thousands [20]), which will provide valuable evidence for understanding and preventing frailty.

This study aims to establish a comprehensive frailty risk prediction model by combining risk factors from socio-demographic, behavioural and social support dimensions, utilizing data from a Chinese national survey of older adults.

## 2. Methods

### 2.1. Samples

This study used data from the Chinese Longitudinal Healthy Longevity Survey (CLHLS) in 2018, covering 23 of 31 provinces in China, which has the largest sample of the oldest-old age group (≥80 years) in the world and a large number of younger older adults as well [21]. CLHLS collected abundant and detailed information on basic information, life evaluation and personality, cognition, behaviour, activities of daily living, personal background, objective examination and illnesses. CLHLS proved to be reliable and reflected the general condition of the Chinese older adults [22]. After excluding the participants with missing data, 14,314 subjects were included in the analyses (Flowchart in Appendix A).

### 2.2. Measures

We treated frailty as a binary outcome indicator and assessed it by the Rockwood Frailty Index (FI), also known as an assessment of defect accumulation. For the CLHLS, the FI proved to be a stable and reliable measure of frailty [23]. In this study, we used nine dimensions to calculate the FI, including cognitive function, chronic illness, ability to perform activities of daily living, activities of daily living, bodily function, self-rated health, hearing ability, visual function, psychological status and other (including heart rhythm, interviewer-rated health status, number of serious illnesses suffered in the past two years) [24]. The FI is the ratio of the defect score to the total score. The higher the FI is, the greater the degree of frailty will be. FI ≥ 0.25 means frailty [25].

The variables included socio-demographic, behavioural, and social support factors (Appendix A). The socio-demographic factors included 11 items: sex, age, nationality, residence, education, marital status, main occupation before age 60, main source of financial support, household income, self-assessed sufficient economic support and self-rated economic level.

The behavioural factors included 35 items: co-residence; staple food; edible oil; main dietary flavour; drinking water; smoking; drinking; exercise; number of times brushing teeth every day; frequency of eating fresh fruit/vegetable/meat/fish/egg/legume/pickle/sugar/garlic/dairy/nut/mushroom or taking algae/vitamin/medicinal plant/tea; taking nutrient supplements (protein/calcium/iron/zinc/compound vitamin/vitamin A or D/DHA/others) usually; and having taken a nutrient supplement or medicine in the past 24 h.

The social support factors included 25 items: number of children; the first person you want to share thoughts with; the first person you ask for help when you have problems; the primary caregiver when ill; ability to access adequate medical service; distance from home to the nearest hospital; the primary payer of medical expenses; having a regular physical examination once every year; having any social security and social insurance (retirement pension/public old-age insurance/private or commercial old-age insurance/public free medical services/urban employee medical insurance/urban resident medical insurance/new rural cooperative medical insurance/commercial medical insurance); and having available community service (personal care/home visit/spiritual comfort and chat/daily shopping/social and recreational activity/legal aid/health education/neighbourhood-relation/others).

### 2.3. Statistical Methods

First, the basic characteristics of the sample were described by mean ± standard deviation for continuous variables and by frequency and percentage for categorical variables. The descriptive analysis mainly explained the older adults’ frailty situation and socio-demographic characteristics.

Next, we used a univariate logistic regression to calculate the unadjusted odds ratios for each of the 71 candidates’ variables. The statistically significant predictors were then checked for multicollinearity. Then, the multivariate logistic regression was performed to determine the most significant predictors from the candidate variables in the univariate analysis and integrate them into the frailty prediction model, which is illustrated by the nomogram.

Finally, we evaluated the effectiveness of the prediction model. First, the receiver-operating characteristic curve (ROC) was used to evaluate the model resolution by calculating forecast probability and the model’s sensitivity and specificity. Then, the calibration curve for the nomogram was validated with bootstrap self-sampling that was repeated 1000 times. The conformity of the predicted and actual probabilities was examined by a calibration curve and Hosmer–Lemeshow test.

All statistical analyses were performed by using SPSS 21.0 (SPSS Inc, Chicago, IL, USA) and R v.4.1.2 (R Foundation for Statistical Computing, Vienna, Austria). All the statistical results were statistically significant as bilateral *p* < 0.05.

## 3. Results

### 3.1. Basic Characteristics of the Sample

Table 1 shows the socio-demographic characteristics of the participants. A total of 14,314 older adults were included in this study, with the FI being 0.175 ± 0.173. The frail group included 3898 older adults (27.2%) aged 85.69 ± 11.76, and the non-frail group included 10,416 older adults (72.8%) aged 85.25 ± 11.69. Moreover, there were 6300 males (44.0%) and 8014 females (56.0%). Most of the older adults were Han Chinese (94.2%); living in urban areas (55.7%); illiterate (49.2%); engaged in agriculture, husbandry, or fishery (61.4%); supported by relatives as their main source of financial support (39.9%); had an annual household income of RMB 1–10 thousand (47.8%); self-assessed with sufficient financial support (86.1%); self-rated their economic level at a moderate level locally (69.7%); and were widowed (57.3%).

### 3.2. Establishment of the Frailty Risk Prediction Model

A total of 71 variables were included in the univariate regression analysis, among which 17 variables (category of drinking water; frequency of taking vitamins; taking calcium/vitamin A or D/DHA/other supplements usually; having taken a nutrient supplement or medicine in the past 24 h; distance from home to the nearest hospital; having public old-age insurance/public free medical services/urban employee medical insurance/urban resident medical insurance/commercial medical insurance; and having available daily shopping/social and recreational activity/legal aid services in community) were not significant (*p*
> 0.05). A total of 54 variables were significant and were included in the subsequent analysis. The tolerances were >0.178 and the variance inflation factors (VIF) were <5.609, indicating that there was no multicollinearity among these variables. 

Only 30 variables, i.e., age; nationality; residence; education; occupation; financial support; self-assessed sufficient economic support; self-rated economic level; marital status; co-residence; staple food; amount of staple food per day; edible oil; main dietary flavour; frequency of taking vegetable/egg/garlic/dairy/nut/tea; main source of water; smoking; drinking; exercise; brushing teeth; the first person you want to share thoughts with; the primary caregiver when ill; the primary payer of medical expense; access to adequate medical service; and regular physical examination, were significant in the multivariate logistic regression and were then combined to build the frailty risk prediction model (Table 2).

Participants aged 70–99 years (OR = 3.03, 95% CI 2.17–4.25) and over 100 years (OR = 9.47, 95% CI 6.64–13.50) had a higher frailty risk than those aged 50–69 years. Participants who were clerks (OR = 0.69, 95% CI 0.53–0.91); self-employed (OR = 0.60, 95% CI 0.37–0.95); agriculture, husbandry, fishery workers (OR = 0.52, 95% CI 0.39–0.70); and house workers (OR = 0.70, 95% CI 0.50–0.99) had a lower frailty risk than those who were professionals and technicians before 60 years old. Participants who rated themselves as poor (OR = 1.76, 95% CI 1.23–2.54) and very poor (OR = 2.29, 95% CI 1.35–3.86) had a higher frailty risk than those who rated themselves as very rich.

Participants who ate corn (OR = 1.64, 95% CI 1.26–2.14); wheat (OR = 1.28, 95% CI 1.10–1.49); half rice and half flour (OR = 1.44, 95% CI 1.26–1.65); and others (OR = 1.68, 95% CI 1.02–2.78) had a higher frailty risk than those who ate rice as a staple food. Participants who brushed their teeth occasionally (OR = 0.72, 95% CI 0.61–0.84); once (OR = 0.52, 95% CI 0.46–0.60); twice (OR = 0.38, 95% CI 0.32–0.45); and three or more times a day (OR = 0.51, 95% CI 0.40–0.65) had a lower frailty risk than those who did not brush their teeth. Participants who did not exercise regularly (OR = 4.53, 95% CI 3.95–5.20) had a higher frailty risk than those who did.

Participants who talked to their children, sons or daughters in law first (OR = 1.65, 95% CI 1.25–2.16) and those who talked to nobody (OR = 2.09, 95% CI 1.42–3.10) when they needed to share their thoughts had a higher frailty risk than those who talked to their spouse first. Participants without a caregiver (OR = 0.25, 95% CI 0.13–0.47) had a lower frailty risk than those who were cared for by their spouse when they were ill. Participants who did not have a regular physical examination once a year (OR = 1.78, 95% CI 1.61–1.97) had a higher frailty risk than those who did. 

The frailty risk prediction model was presented as a nomogram in Figure 1. Locate a value on each variable axis according to the individual’s characteristics and draw a straight line from that value to the “Points” axis to determine the number of points that are associated with that characteristic. For example, those aged 50–69 years old correspond to “0” on the “Age” axis and 0 points; those aged 70–99 years old correspond to “1” on the “Age” axis and 49 points; and those aged ≥100 years old correspond to “2” on the “Age” axis and 99 points. The points for each characteristic are shown in Appendix A. Finally, sum the points for each variable and mark them on the “Total Points” axis, drawing a straight line down the “Risk of Frailty” axis to obtain the individual frailty risk. In addition, to facilitate understanding and interpretation of the nomogram, we developed a web-based version of the dynamic nomogram, https://lisying.shinyapps.io/DynNomapp/ (accessed on 30 June 2022), which is more clearly presented and greatly enhances the usefulness of the prediction model.

### 3.3. Effectiveness of the Frailty Risk Prediction Model

We performed an ROC analysis to assess the prediction model’s performance (Figure 2). The horizontal coordinate is 1-specificity, and the vertical coordinate is sensitivity. The optimal cut-off value of the nomogram total point is 588 in the ROC curve, considering the maximum Youden index value, and the sensitivity and specificity are 77.3% and 83.1%, respectively. Using this cut-off value, older adults were classified as having a low risk of frailty (total point <588) or a high risk of frailty (total point  ≥588). The area under the curve (AUC) is 0.881 (95% CI 0.875–0.887). Compared with the invalid model (chance line), the difference is statistically significant (*p*
< 0.05). In addition, the χ* value of the Hosmer–Lemeshow test is 6.260 (*p* = 0.618) and the calibration curve is slightly nonlinear, which indicates that the predicted probability is consistent with the actual probability and the model fitting degree is ideal (Figure 3).

## 4. Discussion

This study thoroughly considered the socio-demographic, behavioural and social support factors that were proposed by previous studies that may influence frailty in old adults. A frailty risk prediction model among older adults was established with these factors. Multiple validation methods also showed that the model has sufficient statistical power.

Due to the differences in frailty measurement tools, the prevalence of frailty varied widely across studies (range 4.0–59.1%) [26] and was 27.2% in this study. Our study showed that women had a higher FI, which is in line with other reports that older women are vulnerable and frail [26,27]. However, in the multivariate logistic regression, sex was not significantly related to frailty, which seemed to be explained by the “male-female health-survival paradox”. Women are both frail (because they have a poorer health status) and not frail (because they have a better chance of frailty improvement and a lower mortality risk) [27,28,29]. Professionals, technicians and administrators had a higher frailty risk than clerks, the self-employed, agriculture workers, husbandry workers, fishery workers and house workers [30,31]. This is because brain workers are prone to hyperphagia, stress and poor metabolism [32,33]. Soldiers had the highest frailty risk, indicating that the physical and mental damage caused by war can have long-term adverse effects on a person [34]. Those whose main financial source came from local government or community/working by self/relatives had a lower frailty risk than those whose main financial source was retirement wages. Compared to retirement, working can, to some extent, strengthen the older adults’ ties with the outside world and expand the size of their social network, which is conducive to alleviating frailty [35]. Moreover, having financial support from the local government/community or relatives means good social support, which positively influences health. Therefore, when analysing the impact of the economy on frailty, older adults’ subjective satisfaction with the condition of the economy and the financial support resources that are available might be more valuable than their amount of income [36]. Older adults living with household members had a higher frailty risk than those living alone, but a lower risk than those living in nursing homes. This may be because living alone means a higher level of independent living ability and a lower level of frailty [37,38].

For behavioural factors, the main risk factors of frailty were eating habits. Older adults taking rice as their staple food had a lower frailty risk. Dietary and geographical reasons can explain this. In China, the frailty level in the northern region with corn and wheat as the staple food is higher than that in the southern region with rice as the staple food [39]. Participants who ate lard had a lower frailty risk than those who ate vegetable oil. Mass media has advocated for reducing saturated fats intake to prevent obesity; however, this may be inappropriate for frail older adults who are malnourished. Compared with those who preferred insipid-tasting food, those who loved spicy food had a lower frailty risk. Capsaicin has been proven to boost metabolism, reduce fat accumulation, and have anti-inflammatory properties [40]. Eggs and dairy products are essential nutrients for the body and are good sources of high-quality protein and vitamins; however, their recommended daily intake has long been a topic of intense debate, especially for older adults [41,42]. According to our findings, older adults who ate eggs daily had a higher frailty risk than those who ate them occasionally, but had a lower frailty risk than those who never ate eggs. Previous studies showed that an increased intake of eggs, which are rich in cholesterol, was positively associated with cardiovascular disease and cancer mortality in older adults [43]. This is in accordance with our finding. However, further studies are needed to understand the causal relationship between egg intake and frailty. The effects of dairy products on health are also complex and contradictory [44,45]. According to Chinese dietary patterns, dairy products are not as popular among Chinese older adults as tea [46]. In this study, those who drink milk daily had a higher frailty risk. This might be explained by the possibility that frail older adults choose to consume more dairy products to strengthen their bodies and improve their health. Those who eat garlic daily had a lower frailty risk than those who never ate it. Garlic has been proved to have an anti-osteoporotic and anti-ageing effect [47]. Older adults who drank well and spring water had a lower frailty risk than those who drink tap water and lake water. This may be due to the fact that well and spring water is rich in minerals and trace elements that are beneficial to health, such as bicarbonate, calcium sulphate and magnesium [48]. Smoking and drinking are often considered to be risk factors for many diseases, but in this paper, both smoking and drinking were associated with a lower frailty risk. This might be due to the possibility that healthier older adults found it more difficult to stop smoking and drinking. For those who were weaker or suffered from multiple chronic conditions, quitting smoking and drinking might be inevitable due to subjective will and physiological needs. In addition, studies showed that limited smoking can appropriately relieve stress [49], and drinking moderately can benefit the cardiovascular system and alleviate frailty [50,51]. However, measurements of both intakes were absent in this paper, so the results did not suggest that smoking and drinking benefit health.

Social support is very important to cope with frailty. When older adults need to share their thoughts, those who talk first to their spouse enjoy the lowest frailty risk, followed by those who talk with others (including friends, other relatives, and social workers) and their children, while those who have nobody to talk with have the highest frailty risk. This means that social support is associated with a significantly lower frailty risk among older adults, and that spouses are the best supporters, highlighting their critical role in providing informal social support [52]. The older adults who talked with friends had a lower frailty risk than those who talked with children, which may be due to the difficulties in inter-generational communication between parents and children [53]. However, when older adults were ill, those without care had the lowest frailty risk, followed by those who were cared for by their spouse and children; those cared for by others had the highest frail risk. We suggested that this results from survey bias, i.e., older adults with a greater degree of frailty and without caregivers may find it difficult to participate in or complete surveys. In addition, although children are not better communicators than friends, they are better caregivers. This is in line with the reality in China. Participants who did not have money or pay medical expenses by themselves had a lower frailty risk than those who paid by their urban employee or resident medical insurance. The latent factor is that when they utilize insurance to cover their medical expenses, they may suffer from severe disease.

Our research has three strengths. First, we included 71 potential risk factors from socio-demographic, behavioural, and social dimensions, which enabled us to analyse the integrated impact of multiple risk factors on frailty. Second, those risk factors are directly related to the daily life of older adults and are modifiable. Our results could provide important targets for the frailty prevention program. Third, we utilized data from the CLHLS, which has a large sample size and could accurately represent the condition of general Chinese older adults. This study has two limitations. Firstly, causal relationships between risk factors and frailty cannot be established due to the cross-sectional nature of this study. Secondly, research surveys inevitably have missing values. Although this study described the missing values and pre-processed the data using a combination of imputation and deletion methods, the interpretation and promotion of the data analysis results still need to be cautious.

## 5. Conclusions

The present study proposes a comprehensive, feasible and appropriate frailty risk prediction model for Chinese older adults. It is the first frailty risk prediction model based on concurrently socio-demographic, behavioural and social support factors, providing valuable information and targets for the design of frailty prevention programs. 

## Figures and Tables

**Figure 1 ijerph-19-08410-f001:**
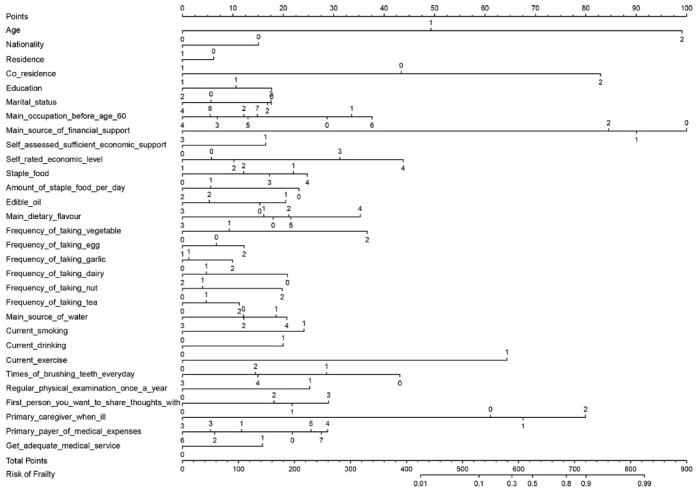
A nomogram to predict frailty risk of older adults.

**Figure 2 ijerph-19-08410-f002:**
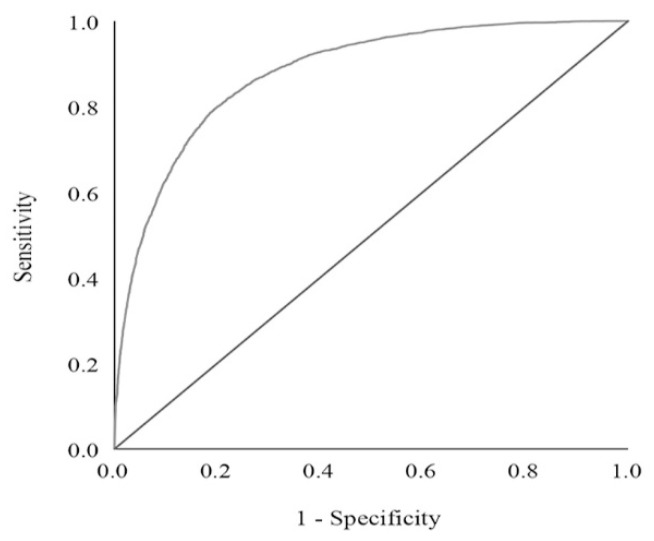
ROC curve of the frailty risk prediction model.

**Figure 3 ijerph-19-08410-f003:**
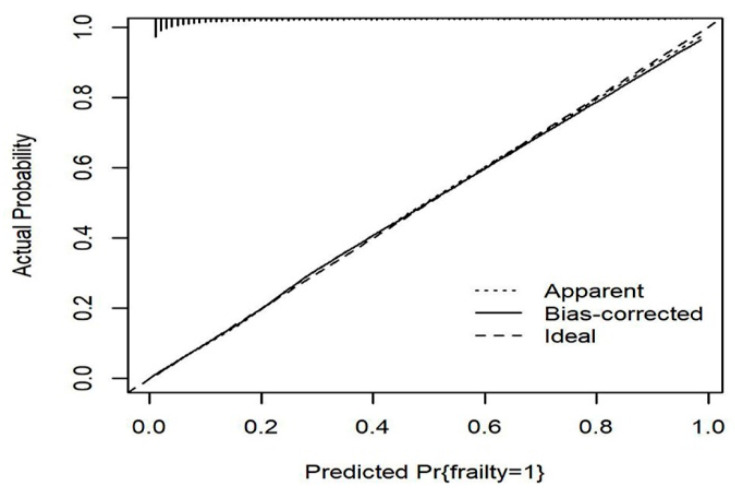
Calibration curve for the established nomogram.

**Table 1 ijerph-19-08410-t001:** Basic characteristics of the sample.

Characteristics	Total	Non-Frail	Frail	FI
Sex				
Man	6300 (44.0)	5080 (48.8)	1220 (31.3)	0.139 ± 0.154
Woman	8014 (56.0)	5336 (51.2)	2678 (68.7)	0.203 ± 0.182
Age (year)				
50–69	1494 (10.5)	1453 (13.9)	41 (1.1)	0.062 ± 0.077
70–99	10,289 (71.9)	8131 (78.1)	2158 (55.4)	0.151 ± 0.154
≥100	2531 (17.7)	832 (8.0)	1699 (43.6)	0.337 ± 0.182
Nationality				
Han	11,611 (94.2)	8295 (93.8)	3316 (95.4)	0.181 ± 0.175
Minority	712 (5.8)	553 (6.3)	159 (4.6)	0.153 ± 0.162
Residence				
Urban	7970 (55.7)	5646 (54.2)	2324 (59.6)	0.185 ± 0.180
Rural	6344 (44.3)	4770 (45.8)	1574 (40.4)	0.162 ± 0.162
Education (year)				
0	6003 (49.2)	3646 (41.6)	2357 (69.0)	0.226 ± 0.185
1–6	3868 (31.7)	3154 (36.0)	714 (20.9)	0.136 ± 0.152
>6	2319 (19.0)	1972 (22.5)	347 (10.2)	0.126 ± 0.145
Marital status				
Currently married and living with spouse	5651 (39.9)	5051 (48.9)	600 (15.6)	0.104 ± 0.122
Separated	241 (1.7)	196 (1.9)	45 (1.2)	0.128 ± 0.151
Divorced	45 (0.3)	33 (0.3)	12 (0.3)	0.172 ± 0.154
Widowed	8120 (57.3)	4944 (47.9)	3176 (82.3)	0.226 ± 0.186
Never married	122 (0.9)	97 (0.9)	25 (0.6)	0.155 ± 0.146
Main occupation before age 60				
Professionals and technician	817 (6.7)	634 (7.3)	183 (5.3)	0.156 ± 0.166
Administrative manager	498 (4.1)	363 (4.2)	135 (3.9)	0.176 ± 0.183
Clerk	1794 (14.7)	1323 (15.1)	471 (13.7)	0.182 ± 0.183
Self-employed	233 (1.9)	171 (2.0)	62 (1.8)	0.168 ± 0.163
Agriculture/husbandry/fishery	7478 (61.4)	5418 (62.0)	2060 (60.1)	0.174 ± 0.168
House worker	824 (6.8)	492 (5.6)	332 (9.7)	0.229 ± 0.192
Soldier	111 (0.9)	81 (0.9)	30 (0.9)	0.175 ± 0.178
Never worked	196 (1.6)	99 (1.1)	97 (2.8)	0.282 ± 0.221
Others	222 (1.8)	163 (1.9)	59 (1.7)	0.168 ± 0.174
Main source of financial support				
Retirement wages	3546 (22.3)	2697 (28.3)	849 (23.5)	0.167 ± 0.174
Relative (s)	6995 (53.3)	4677 (49.1)	2318 (64.1)	0.198 ± 0.178
Local government or community	1399 (8.8)	963 (10.1)	436 (12.1)	0.196 ± 0.171
Work by self	1195 (7.5)	1182 (12.4)	13 (0.4)	0.053 ± 0.056
Household income (10,000 RMB)				
≤0.1	736 (5.6)	507 (5.3)	229 (6.4)	0.196 ± 0.173
0.1–0.3	965 (7.4)	742 (7.8)	223 (6.3)	0.157 ± 0.151
0.3–0.8	1707 (13.0)	1251 (13.1)	456 (12.8)	0.170 ± 0.171
0.8–1.0	956 (7.3)	691 (7.2)	265 (7.4)	0.173 ± 0.170
1.0–10.0	6259 (47.8)	4524 (47.4)	1735 (48.7)	0.175 ± 0.177
>10.0	2474 (18.9)	1821 (19.1)	653 (18.3)	0.175 ± 0.174
Self-assessed sufficient economic support				
Yes	12,240 (86.1)	9058 (87.5)	3182 (82.3)	0.168 ± 0.169
No	1982 (13.9)	1297 (12.5)	685 (17.7)	0.214 ± 0.189
Self-rated economic level				
Very rich	371 (2.6)	280 (2.7)	91 (2.4)	0.160 ± 0.172
Rich	2418 (17.1)	1890 (18.3)	528 (13.8)	0.150 ± 0.158
So-so	9877 (69.7)	7232 (70.0)	2709 (69.0)	0.173 ± 0.172
Poor	1311 (9.3)	833 (8.1)	478 (12.5)	0.223 ± 0.189
Very poor	191 (1.3)	99 (1.0)	92 (2.4)	0.270 ± 0.199

Notes: N = 14,314 in Total; 10,416 in Non-frail; 3898 in Frail.

**Table 2 ijerph-19-08410-t002:** Univariate and multivariate logistic regressions of the influencing factors for frailty.

Variables	OR (95% CI)	AOR (95% CI)	β
Age (year) (*n* = 14,314, 100%) (reference: 50–69)			
70–99	9.41 (6.87–12.88) **	3.10 (2.22–4.34) **	1.110
≥100	72.37 (52.48–99.79) **	9.76 (6.86–13.90) **	2.248
Nationality (*n* = 12,323, 86.1%) (reference: Han)			
Minority	0.70 (0.59–0.83) **	0.71 (0.57–0.87) *	−0.345
Residence (*n* = 14,314, 100%) (reference: Urban)			
Rural	0.90 (0.86–0.93) **	0.87 (0.78–0.97) *	−0.115
Education (*n* = 12,190, 85.2%) (year) (reference: 0)			
1–6	0.41 (0.38–0.45) **	0.85 (0.76–0.95) *	−0.194
>6	0.31 (0.28–0.35) **	0.67 (0.55–0.81) **	−0.460
Marital status (*n* = 14,179, 99.1%) (reference: Currently married and living with spouse)			
Separated	1.90 (1.36–2.66) **	1.32 (0.86–2.02)	
Divorced	2.92 (1.88–4.52) **	1.29 (0.73–2.30)	
Widowed	5.34 (4.86–5.88) **	1.31 (1.07–1.60) *	0.271
Never married	2.15 (1.37–3.36) *	0.90 (0.48–1.67)	
Main occupation before age 60 (*n* = 12,173, 85.0%) (reference: Professional and technician)			
Administrative manager	1.24 (0.96–1.59)	1.09 (0.78–1.53)	
Clerk	1.20 (0.99–1.46) *	0.69 (0.53–0.91) *	−0.371
Self-employed	0.96 (0.74–1.23)	0.60 (0.37–0.95) *	−0.517
Agriculture, husbandry, fishery	1.31 (1.11–1.56) *	0.52 (0.39–0.70) **	−0.645
House worker	2.02 (1.64–2.49) **	0.70 (0.50–0.99) *	−0.354
Soldier	1.15 (0.78–1.68)	1.19 (0.65–2.18)	
Never worked	3.20 (2.33–4.39) **	0.72 (0.45–1.14)	
Others	1.26 (0.90–1.77)	0.58 (0.35–0.93) *	−0.553
Main source of financial support (*n* = 13,135, 91.8%) (reference: Retirement wages)			
Local government or community	1.37 (1.20–1.56) **	0.82 (0.68–0.98) *	−0.202
Relative(s)	1.46 (1.34–1.60) **	0.70 (0.56–0.89) *	−0.352
Work by self	0.05 (0.03–0.08) **	0.10 (0.06–0.17) **	−2.268
Self-assessed sufficient economic support (*n* = 14,222, 99.4%) (reference: Yes)			
No	1.50 (1.36–1.66) **	1.45 (1.24–1.69) **	0.370
Self-rated economic level (*n* = 14,168, 99.0%) (reference: Very rich)			
Rich	0.86 (0.67–1.11)	0.85 (0.61–1.19)	
So-so	1.14 (0.90–1.45)	1.09 (0.79–1.51)	
Poor	1.77 (1.36–2.29) **	1.76 (1.23–2.54) *	0.567
Very poor	2.86 (1.98–4.14) **	2.29 (1.35–3.86) *	0.826
Co-residence (*n* = 14,132, 98.7%) (reference: With household member(s))			
Alone	0.50 (0.44–0.56) **	0.38 (0.33–0.44) **	−0.975
In an institution	3.29 (2.74–3.94) **	2.49 (1.93–3.22) **	0.914
Staple food (*n* = 14,285, 99.8%) (reference: Rice)			
Corn (maize)	1.42 (1.17–1.71) **	1.64 (1.26–2.14) **	0.495
Wheat (noodles and bread, etc.)	1.37 (1.24–1.51) **	1.28 (1.10–1.49) *	0.249
Half rice and half flour	1.47 (1.34–1.62) **	1.44 (1.26–1.65) **	0.366
Others	4.60 (3.22–6.57) **	1.68 (1.02–2.78) *	0.520
Amount of staple food per day (n = 14,233, 99.4%) (kg) (reference: <0.2)			
0.2–0.5	0.45 (0.41–0.48) **	0.68 (0.61–0.76) **	−0.384
>0.5	0.30 (0.25–0.36) **	0.59 (0.46–0.76) **	−0.526
Edible oil (*n* = 14,269, 99.7%) (reference: Other vegetable oils)			
Gingili grease	1.37 (0.90–2.07)	1.08 (0.61–1.92)	
Lard	0.75 (0.66–0.86) **	0.78 (0.66–0.93) *	−0.246
Other animal’s fat	1.04 (0.60–1.81)	0.69 (0.33–1.44)	
Main dietary flavour (*n* = 14,269, 99.7%) (reference: Insipidity)			
Salty	0.79 (0.71–0.87) **	0.97 (0.85–1.10)	
Sweet	1.53 (1.31–1.790 **	1.04 (0.84–1.29)	
Hot	0.40 (0.29–0.56) **	0.67 (0.45–0.98) *	−0.400
Crude	1.52 (0.60–3.87)	1.49 (0.45–4.91)	
Others	1.29 (1.01–1.55)	1.08 (0.86–1.38)	
Frequency of taking vegetables (*n* = 14,283, 99.8%) (reference: Almost every day)			
Occasionally	1.52 (1.40–1.64) **	1.28 (1.15–1.43) **	0.246
Rarely or never	5.74 (4.78–6.90) **	2.19 (1.71–2.80) **	0.784
Frequency of taking egg (*n* = 14,182, 99.1%) (reference: Almost every day)			
Occasionally	0.73 (0.67–0.78) **	0.88 (0.78–0.99) *	−0.127
Rarely or never	0.97 (0.85–1.11)	1.05 (0.87–1.27)	
Frequency of taking garlic (*n* = 14,175, 99.0%) (reference: Almost every day)			
Occasionally	1.33 (1.20–1.48) **	1.03 (0.90–1.19)	
Rarely or never	2.48 (2.21–2.78) **	1.20 (1.03–1.41) *	0.184
Frequency of taking dairy (*n* = 14,163, 98.9%) (reference: Almost every day)			
Occasionally	0.66 (0.60–0.72) **	0.72 (0.63–0.82) **	−0.332
Rarely or never	0.68 (0.62–0.74) **	0.63 (0.55–0.73) **	−0.455
Frequency of taking nut (*n* = 14,161, 98.9%) (reference: Almost every day)			
Occasionally	1.04 (0.86–1.25)	1.11 (0.86–1.44)	
Rarely or never	2.45 (2.04–2.94) **	1.56 (1.20–2.01) *	0.442
Frequency of taking tea (*n* = 14,011, 97.9%) (reference: Almost every day)			
Occasionally	1.43 (1.20–1.71) **	1.16 (0.93–1.46)	
Rarely or never	2.42 (2.14–2.72) **	1.32 (1.12–1.54) *	0.274
Main source of water (*n* = 14,060, 98.2%) (reference: From a well)			
From a river or lake	1.00 (0.68–1.48)	1.18 (0.73–1.91)	
From a spring	0.80 (0.61–1.05)	1.00 (0.71–1.41)	
From a pond or pool	0.68 (0.26–1.80)	0.80 (0.26–2.45)	
Tap water	1.23 (1.12–1.36) **	1.22 (1.07–1.40) *	0.200
Current smoking (*n* = 14,174, 99.0%) (reference: Yes)			
No	3.05 (2.66–3.49) **	1.80 (1.51–2.15) **	0.590
Current drinking (*n* = 14,103, 98.5%) (reference: Yes)			
No	2.87 (2.51–3.29) **	1.60 (1.34–1.91) **	0.470
Current exercise (*n* = 14,127, 98.7%) (reference: Yes)			
No	5.93 (5.28–6.65) **	4.53 (3.95–5.20) **	1.511
Number of times brushing teeth everyday (*n* = 13,952, 97.5%) (reference: Do not brush)			
Occasionally	0.47 (0.41–0.53) **	0.72 (0.61–0.84) **	−0.329
Once	0.27 (0.26–0.30) **	0.52 (0.46–0.60) **	−0.645
Twice	0.20 (0.18–0.23) **	0.38 (0.32–0.45) **	−0.977
Three or more times	0.30 (0.25–0.36) **	0.51 (0.40–0.65) **	−0.669
Regular physical examination once a year (*n* = 12,043, 84.1%) (reference: Yes)			
No	3.21 (2.97–3.46) **	1.78 (1.61–1.97) **	0.574
First person you want to share thoughts with (*n* = 13,952, 97.5%) (reference: Spouse)			
Children, sons in law or daughters in law	5.99 (5.36–6.68) **	1.65 (1.25–2.16) **	0.498
Others	3.91 (3.25–4.70) **	1.48 (1.02–2.15) *	0.395
Nobody	6.15 (4.97–7.63) **	2.09 (1.42–3.10) **	0.739
Primary caregiver when ill (*n* = 14,086, 98.4%) (reference: Spouse)			
Children, sons in law or daughters in law	5.55 (4.93–6.25) **	1.16 (0.94–1.42)	
Others	10.37 (8.79–12.23) **	1.52 (1.15–1.99) *	0.415
Nobody	0.65 (0.37–1.12)	0.25 (0.13–0.47) **	−1.405
Primary payer of medical expenses (*n* = 13,841, 96.7%) (reference: Urban employee/resident medical insurance)			
Cooperative medical scheme	0.82 (0.74–0.91) **	0.83 (0.68–1.02)	
Private medical insurance	0.90 (0.58–1.43)	0.74 (0.41–1.34)	
Self	0.49 (0.43–0.57) **	0.70 (0.57–0.86) *	−0.360
Spouse	0.63 (0.44–0.89) *	1.21 (0.75–1.94)	
Children	1.80 (1.62–1.99) **	1.11 (0.91–1.34)	
No money to pay	1.90 (0.81–4.49)	0.61 (0.20–1.83)	
Others	1.98 (1.50–2.61) **	1.08 (0.74–1.59)	
Access to adequate medical service (*n* = 14,192, 99.1%) (reference: Yes)			
No	2.16 (1.78–2.62) **	1.42 (1.09–1.87) *	0.353

Notes: * *p*
<0.05; ** *p*
< 0.001; AOR = adjusted odd ratio.

## Data Availability

Restrictions apply to the availability of these data. Data were obtained from the Research Ethics Committee of Peking University and available at the Chinese Longitudinal Healthy Longevity Survey (CLHLS)—Duke Aging Center. The Chinese Longitudinal Healthy Longevity Survey (CLHLS) data are publicly available from Peking University. Researchers can obtain these data after submitting a data use agreement to the CLHLS team.

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
