# Peer review of "Frailty Risk Prediction Model among Older Adults: A Chinese Nation-Wide Cross-Sectional Study"

_ijerph, 2022, doi:10.3390/ijerph19148410_

Round 1
Reviewer 1 Report
This manuscript examined the frailty risk prediction model in older Chinese by finding the impact of socio-demographic, behavioral, and social support factors on the risk of developing frailty. The topic of the study is interesting, and the study results might have a potential clinical impact. However, this manuscript needs major revision and has to address the comments listed below: Major comments:
Introduction/Background:Page 2, 49-50 line: The authors stated that “Moreover, most of these studies were carried out in developed countries”. The authors should provide what previous studies are (i.e., developed countries’ studies) and add the references. Also, the authors should provide a brief rationale for why the results from developed and non-developed countries are different.
Page 2, 51-52 line: The authors stated that “especially among Chinese older adults which is the largest aging population in the world”. Please provide more information and reference(s).
Page 2, 57 lines: Please provide the study hypotheses.This Introduction needs revision by considering the comments and questions above.Methods:Page 2, 65 lines: This reviewer strongly recommends the authors provide the flow chart of participants (e.g., cohort diagram).
Page 2, 77-92 lines: Please briefly address why the 35 items as behavioral factors and 25 items as social support factors are the risk of developing frailty. Also, please provide the reference(s).
Results:Page 3, 117-118 lines: Please provide the mean and standard deviation of the age for each group (non-frail and frail).
Page 4, 137 lines: Please provide more explanation about Figure 1.
Page 4, 138 lines: The authors do not provide information about how many missing data (i.e., behavioral and social support factors) were for Table 2. Please provide the information using the frequency and %.
Page 7, 167-168 lines: Please provide more explanation about Figure 2 (i.e., what the dependent variable is and what the independent variables are).
Discussion:
Page 8, 183 lines: Please clarify the “eating patterns”.
Page 10, 254 lines: The authors stated that “Regarding the limitations of this research”. However, the authors do not provide the limitations. Previous sentences were about 3 strengths.
Page 8, 258 lines: The authors stated that “What’s more”, but this sentence is grammatically incomplete.
By considering the comments and suggestions above, this section should be revised extensively.
Author Response
Introduction
- Thank you very much for your suggestions. We added studies from some developed countries and their references, and then explained why the results are different for developed and developing countries. The series of additions help us to articulate more clearly the importance of this study. The statement in the text is as follows: In addition, most frailty prediction models were carried out in developed countries[8]. For example, a study from Tokyo developed a frailty predictive model based on socio-demographic, medical, behavioural and subjective factors(7). There is also a study in French that developed the FRAGIRE tool to assess the risk of frailty in older adults(8). As the great differences in terms of socio-demographic characteristics, behaviour and social environment between developed and developing countries, the frailty prediction models will also differ significantly[9, 10]. Still unknown is how socio-demographic, behavioural, and social support factors would influence and predict frailty together in developing countries. Hence, there is a need to establish a frailty risk prediction model for Chinese older adults, the largest ageing population in the world, which will provide valuable evidence for understanding frailty. See page 2, 49-59 lines.
-
Thank you for the suggestion. According to the “World Population Ageing 2019: Highlights”, China has the largest population of older adults in the world with approximately 164,487 thousand older adults in 2019. See page 2, 56-57 lines.
-
Thank you very much for your suggestion. Our study hypothesis is stated as follows, for older adults, socio-demographic, behavioural and social support factors were significantly associated with frailty in an integrated manner. See page 2, 58-60 lines.
Methods:
- Thank you for your reminder. We do appreciate it. We added the flowchart in Supplementary Figure S1. See page 2, 73 lines.
-
Thank you for your suggestion. We chose these variables because they were proven to be associated with frailty in previous research. We added more specific descriptions and their references in the manuscript. See page 2, 40-46 lines.
Results
-
Thank you for your suggestion. The frail group included 3,898 older adults (27.2%) with the mean age and standard deviation being 85.6911.76 and the non-frail group included 10,416 older adults (72.8%) with the age being 85.2511.69. See page 3, 124-126 lines.
-
Thank you for the reminder. We added an explanation for Figure 1 in the manuscript as follows, the frailty risk prediction model was presented as a nomogram in Figure 1. For example, locate the Age and draw a line straight upward to the “Points” axis to determine the score associated with that age group. Repeat this process for each variable and add the scores of each covariate to obtain the “Total Points”, and draw a line straight down to the “Risk of Frailty” axis to determine the frailty probability. The optimal cutoff value of the nomogram total score was 588 in the ROC curve considering the maximum Youden index value, and the sensitivity and specificity were 77.3% and 83.1%, respectively. Using this cutoff value, older adults with total nomogram scores of <588 points or ≥588 points were classified as having a low or high risk of frailty. See page 5, 159-162 lines and page 8, 194-198 lines.
-
Thank you so much for this suggestion. We added missing data to the text. See Table 2.
-
Thank you for your suggestion. We added detailed information in Figure 2 in the manuscript. The horizontal coordinate is 1-specificity and the vertical coordinate is sensitivity. See page 8, 193 lines.
Discussion
-
Thank you for the comment. The “eating patterns” refers to a series of variables that describe the eating habits of older adults, such as the frequency of eating vegetable/ egg/ garlic/ dairy/ nut/ tea. We supplemented the “eating patterns” with corresponding specific variables as follows, the final results show that socio-demographic factors including age, nationality, residence, education, occupation, financial support, self-assessed sufficient economic support, self-rated economic level, marital status, behavioural factors including co-residence, staple food, amount of staple food per day, edible oil, main dietary flavour, frequency of taking vegetable/ egg/ garlic/ dairy/ nut/ tea, main source of water, smoking, drinking, exercise, brushing teeth, social support factors including regular physical examination, the first person you want to share thoughts with, primary caregiver when ill, get adequate medical service and the primary payer of medical expense were the independent predictors of frailty. A frailty risk prediction model among older adults was established with these factors. See page9, 207-218 lines.
-
Thank you for the correction. We added the limitations in the Discussion as follows, this study has two limitations. Firstly, causal relationships between risk factors and frailty were not able to be established due to the cross-sectional nature of this study. Secondly, research surveys inevitably have missing values. Although this study describes missing values and preprocesses the data using a combination of imputation and delete methods, the interpretation and promotion of data analysis results still need to be cautious. See page 10, 287-291 lines.
-
Thank you for reminding. We rewrote the conclusion as follow, The present study proposed a comprehensive, feasible and appropriate frailty risk prediction model for Chinese older adults. It is the first frailty risk prediction model that is based on socio-demographic, behavioural and social factors concurrently, providing valuable information and targets for the design of frailty prevention programs. See page 10, 293-296 lines.

Reviewer 2 Report
This study used national survey to build frailty prediction model. It is an important question and here are some comments:
1. P2 line 47, the authors stated ‘However, rare study explore the integrated impact of those risk factors on the development of frailty’. However, this is cross-sectional study and the concurrent status of frailty (yes vs no) was used as dependent variable, it cannot detect the risk factors of the development of frailty.
2. P2 line 71: please explain what is ‘judgement value’ of FI?
3. Please provide the ranges and meaning of the FI scores, for example high FI score means worse frailty? In the results, it seemed to use categorical variables into frail vs non-frail, and what is the cut-point?
4. P2 line 77: what is ‘main flavor’? What is ‘living status’ and what is the difference between ‘residence’ and living status? Some measurements did not have clear definition, for example psychological consulting and social recreation in community services?
5. P3 how were the behavioral and social support factors entered in the univariate and multi-variate analyses? Especially many were categorical variables.
6. Furthermore, how was the prediction risk model built? More descriptions about statistical analysis are needed.
7. What is the model? And how to integrate the significant variables to test the predictive effect of the model? What is the cut-off point of the model? Without these information, researchers or clinical staff could not apply this model in their works.
8. The results found various kinds of significant factors, and the detail explanations about why these factors were significant to predict frailty will be needed in the discussion. However, the contents of the model included too many factors to be used in clinical situation.
Author Response
1. Thank you for your perspicacious comment to improve the preciseness of our paper. We revised this sentence as follows, “However, few studies have examined the combined impact of these risk factors on frailty” and replaced other words in the article that indicate “the development of frailty” accordingly. See page 2, 45lines.
2. Thank you for your comment. We revised the sentence as follows, “FI is the ratio of the defect score to the total score. The higher the FI, the greater the degree of debilitation. FI≥0.25 means frailty”. See page 2, 76-81 lines.
3. Thanks very much for your suggestion. We added contents as follows, “FI is the ratio of the defect score to the total score. The higher the FI, the greater the degree of debilitation. FI≥0.25 defined as frailty.” See page 2, 76-81 lines. In this paper, the meaning and range were 0.173 (0-0.875). See page 3, 127 line.
4. Sorry for the confusion. We added Supplementary Table 1, which explained the coding and categories of the 35 behavioural factors and 25 social support factors. Detailed information about the socio-demographic factors can be found in Table 1.
- The ‘main flavor’means the main flavour of the diet, including salty, sweet, hot and crude. We changed ‘main flavor’ to ‘main dietary flavour’.
- The ‘living status’includes ‘living with household member(s)’, ‘living alone’and ‘living in the institution’. We changed ‘living status’ to ‘co-residence’.
- While ‘residence’ includes ‘living in urban’ and ‘living in rural’. ‘psychological consulting’ means that community workers provide spiritual comfort and chat for older people to relieve boredom, chatting and relaxation. We changed ‘psychological consulting’ to ‘spiritual comfort and chat’.
- ‘social recreation’ means that community workers organize social and recreational activities for older adults. We changed ‘social recreation’ to ‘social and recreational activity’.
5. Thanks for your question. The factors in this paper were entered into univariate and multivariate analyses by the forced entry method. For each categorical variable, reference groups were set. See Table 2.
6. Thanks for your valuable suggestion. In fact, the process of logistic regression is the process of building the predictive model. We re-described the process of building the predictive model as detailed follow. We used a univariate logistic regression model to calculate unadjusted odds ratios for each of the 79 candidate predictor variables. The statistically significant predictors were then checked for multicollinearity and included in the multivariate logistic regression. The statistically significant predictors in the multivariate logistic regression were then re-run in the multivariate logistic regression to obtain the final frailty prediction model, which was illustrated by the nomogram. See page 3, 109-114 lines.
7. Thanks for your comments.
- what the model is? The model is a frailty prediction model for older adults, i.e., with age, nationality, residence, education, occupation, financial support, self-assessed sufficient economic support, self-rated economic level, marital status, co-residence, staple food, amount of staple food per day, edible oil, main dietary flavour, frequency of taking vegetable/ egg/ garlic/ dairy/ nut/ tea, main source of water, smoking, drinking, exercise, brushing teeth, first person you want to share thoughts with, primary caregiver when ill, get adequate medical service, primary payer of medical expense and regular physical examination as the dependent variables and frailty as the independent variable. The frailty risk prediction model was presented as a nomogram in Figure 1. See page 5, 149-160 lines.
- How to integrate the significant variables to test the predictive effect of the model? For example, locate the “Age” and draw a line straight upward to the “Points” axis to determine the score associated with that age group. Repeat this process for each variable and add the scores of each covariate to obtain the “Total Points”, then draw a line straight down to the “Risk of Frailty” axis to determine the frailty probability. To assess the performance of the prediction model, we performed a ROC analysis (Figure 2). The sensitivity and specificity were 77.3% and 83.1%, respectively. The χ2 value of the Hosmer-Lemeshow test was 6.260 (p=0.618), and the calibration curve was slightly nonlinear, which indicated that the predicted probability is consistent with the real probability and the model fitting degree is ideal (Figure 3). See page 5, 158-160 lines, and See page 8, 193-200 lines.
- What is the cut-off point of the model? The optimal cut-off value of the nomogram total score was 588 in the ROC curve considering the maximum Youden index value, and the sensitivity and specificity were 77.3% and 83.1%, respectively. Using this cut-off value, older adults were classified as having low high risk frailty (total point <588) or high risk frailty (total point ≥588). See page 8, 193-196 lines.
8. Thank you for your suggestion. We chose 71 variables as candidate predictor variables which were available from the CLHLS questionnaire and had been proven to be associated with frailty in previous research studies. See 2nd paragraph of the Introduction and the Discussion. Then in this paper, we selected 30 predictors through the process of building a predictive model and further demonstrated the importance of these variables for frailty. We think that although the model has many variables, leading to some limitations in its clinical application. But, these variables encompass socio-demographic aspects, life behavioural aspects and social support aspects, which are extensive, comprehensive, accessible and feasible. It is particularly applicable to community-based screening of frailty in older adults. Therefore, we believe that the frailty prediction model for older adults is of practical value.

Round 2
Reviewer 1 Report
Overall, extensive editing of English is significantly needed.
Author Response
Dear reviewer,
On behalf of my co-authors, we greatly appreciate your warm work on our manuscript entitled “Frailty risk prediction model among older adults, a Chinese nation-wide cross-sectional study” (ID, ijerph-1758533), which contributed largely to improve the quality of our paper. According to your suggestion, we invited a native English speaker to revise this article’s grammar, tenses and syntax carefully. We hope the revised English language and style will be approved, but please do not hesitate to contact us if it still does not meet the requirements or if the manuscript needs other changes.
Once again, thank you very much for your comments and suggestions.

Reviewer 2 Report
Most of comments were answered properly, however here are some comments:
1. The image resolution of Figure 1 is too low to read. In addition, the rules of counting the nomogram total point is also unclear. The readers can not apply the information from Figure to count the total point. For example, what is the score of minority? Please provide the detail rules in Figure or in the manuscript and improve the image resolution.
2. The discussions about the significant predictors needed to be improved, for example, eating an egg (the results was inconsistent with taking dairy), tap water, no smoking, no drinking, first person you want to share thoughts with (children), primary caregiver when ill (nobody).
Author Response
Reviewer comment 1
Thanks very much for your suggestion. We made the following changes,
- We increased the image resolution of Figure 1 to 300dpi, and we will upload the qualified image of Figure 1 separately.
- We described the counting rules for the nomogram in more detail. See page 7 in the last paragraph.
- To facilitate understanding and interpretation of the nomogram, we developed a web-based dynamic nomogram(https://lisying.shinyapps.io/DynNomapp/), which presents the results more clearly and greatly enhances the usefulness of the prediction model.
Reviewer comment 2
Thanks very much for your suggestion. We rewrite the discussions about the significant predictors. See pages 9-11.
